# Protective Effect and Mechanism of Boswellic Acid and Myrrha Sesquiterpenes with Different Proportions of Compatibility on Neuroinflammation by LPS-Induced BV2 Cells Combined with Network Pharmacology

**DOI:** 10.3390/molecules24213946

**Published:** 2019-10-31

**Authors:** Xiao-dong MIAO, Li-jie ZHENG, Zi-zhang ZHAO, Shu-lan SU, Yue ZHU, Jian-ming GUO, Er-xin SHANG, Da-wei QIAN, Jin-ao DUAN

**Affiliations:** 1Jiangsu Key Laboratory for High Technology Research of TCM Formulae; Jiangsu Collaborative Innovation Center of Chinese Medicinal Resources Industrialization; National and Local Collaborative Engineering Center of Chinese Medicinal Resources Industrialization and Formulae Innovative Medicine, Nanjing University of Chinese Medicine, Nanjing 210023, China; miaoxiaodong94111@163.com (X.-d.M.); 20181402@njucm.edu.cn (Z.-z.Z.); zhuyue@njucm.edu.cn (Y.Z.); njuguo@njucm.edu.cn (J.-m.G.); shex@njucm.edu.cn (E.-x.S.); qiandwnj@126.com (D.-w.Q.); 2Jiangsu Key Laboratory of Research and Development in Marine Bio-resource Pharmaceutics, Nanjing University of Chinese Medicine, Nanjing 210023, China; 18851003108@163.com

**Keywords:** frankincense, myrrha, neuroinflammation, boswellic acid, myrrha sesquiterpenes, different proportions compatibility, BV2 cells, network pharmacology

## Abstract

Frankincense and myrrha (FM), commonly used as a classical herbal pair, have a wide range of clinical applications and definite anti-inflammatory activity. However, anti-neuroinflammation effects and mechanisms are not clear. In this study, we adopted a lipopolysaccharide (LPS)-induced microglial (BV2) cell model and a network pharmacology method to reveal the anti-neuroinflammatory effects and mechanisms of boswellic acid (BA) and myrrha sesquiterpenes (MS) with different proportions of compatibility. The data showed that the different ratios of BA and MS had different degrees of inhibition of interleukin-1β (IL-1β), IL-6, and inducible nitric oxide synthase (iNOS) mRNA expression, down-regulated the phosphor-nuclear factor kappa B/nuclear factor kappa B (p-NF-ҡB)/(NF-ҡB), phosphorylated protein kinase b/protein kinase b (p-AKT/AKT), and Toll-like receptor 4 (TLR4) protein expression levels, and increased phospho-PI3 kinase (p-PI3K) protein expression levels. When the ratios of BA and MS were 10:1, 5:1, and 20:1, better effective efficacy was exhibited. According to the correlation analysis between the effect index and bioactive substances, it was suggested that 2-methoxy-5-acetoxy -fruranogermacr-1(10)-en-6-one (Compound 1), 3*α*-acetyloxylanosta-8,24-dien-21-oic acid (Compound 2), 11-keto-boswellic acid (Compound 3), and 3-acetyl-11-keto-*β* -boswellic acid (Compound 4) made important contributions to the treatment of neuroinflammation. Furthermore, based on the network pharmacological analysis, it was found that these four active compounds acted on 31 targets related to neuroinflammation and were involved in 32 signaling pathways which mainly related to the immune system, cardiovascular system, and nervous system, suggesting that BA and MS could be used to treat neuroinflammation.

## 1. Introduction

Neuroinflammation widely occurs in the pathogenesis of various central nervous system (CNS) diseases, such as ischemic stroke, traumatic brain injury, Parkinson’s disease, and Alzheimer’s disease [1]. Neuroinflammation is an immune response activated by microglia and astrocytes in the CNS, which results in the release of pro-inflammatory cytokines and chemokines [2]. Microglia play an important role in the response of the CNS, with neuromodulation, neurotrophic, and neuroimmune functions. When the central nervous system is damaged, microglia are activated to migrate to the damaged site, releasing a series of pro-inflammatory, anti-inflammatory, and neurotoxic reactions [3,4,5]. Therefore, inhibition of microglial activation may be an effective strategy for the treatment of neuroinflammation. Lipopolysaccharide (LPS) is an endotoxin that induces inflammation and is often used to stimulate microglia to construct a useful in vitro model for studying the mechanisms of neuronal damage [6]. In recent years, there have been many reports on the establishment of neuroinflammatory models by LPS-activated microglia [7,8,9]. Currently, non-steroidal anti-inflammatory drugs (NSAIDs) are widely used to treat inflammation and other related diseases, but due to the obvious adverse reactions caused by long-term use they are limited in clinical applications. Therefore, it is necessary to find a new drug with few or no side effects [10].

Frankincense is a gelatinous resin exuded from *Boswellia carterii* Birdw. or *Boswellia bhaw-dajiana* Birdw. Boswellic acid (BA) is the main bioactive compound in frankincense, with outstanding anti-inflammatory properties [11]. It has been found that frankincense extract can be used in the treatment of systemic inflammation induced by LPS in mice [12]. Myrrha is an oily, gelatinous substance exuded from the bark of *Commiphora myrrha* Engl. or *Commiphora mo1mol* Engl., and is recorded in “Kai Bao Materia Medica”. Its main bioactive ingredients are myrrha sesquiterpenes (MS), which have cytotoxic, antibacterial, anti-inflammatory, analgesic, and anti-oxidant effects. It was found that myrrha terpene derivatives could inhibit the activity of the cyclooxygenase 1 (COX-1) enzyme to a certain extent [13,14,15].

Frankincense and myrrha (FM) represent a clinically typical herbal pairing found in the “Ruxiang Zhitong San”, with effects of invigorating circulation, reducing stasis, astringing wounds, and generating muscle. Modern pharmacological studies have shown that FM have obvious anti-inflammatory effects [16]. The FM extract can exert anti-positive T cell cytotoxic cells (CD^8+^ T cell)-mediated liver cancer activity [17]. However, the combination of BA and MS in the treatment of neuroinflammation has not been reported.

Network pharmacology, a systematic and effective tool for the study of therapeutic mechanisms of traditional Chinese medicine, is established by constructing of component-targets, target-pathways, and target-diseases based on existing databases [18], and helping us to understand the integration synergistic mechanism of Chinese medicine formula at the molecular network level [19,20].

In our study, the neuroinflammatory model of BV2 cells induced by LPS was established to evaluate anti-neuroinflammatory effects, and chemical ingredients were determined by UPLC-TQ/MS with different ratios of BA and MS. Then, the key chemical composition of the contribution was obtained by the Pearson correlation analysis. On this basis, through network pharmacology and with the help of multiple online database resources, a chemical–target–pathway network diagram was constructed to predict the key targets and main signaling pathways, and the molecular mechanisms of BA and MS compatibility with respect to neuroinflammation were constructed, providing the basis for the rational use and in-depth research and development of the new drug.

## 2. Results

### 2.1. Effects of BA and MS on the Viability of BV2 Microglia

The results of MTT assay for the detection of cytotoxicity are shown in Figure 1. When the concentration was greater than 12.5 μg·mL^−1^, the growth of BV2 cells was inhibited to varying degrees, and the inhibition rate was positively correlated with the concentration. Therefore, we selected the concentration of 10 μg·mL^−1^ in this test.

### 2.2. Expression of IL-1β, IL-6, and iNOS mRNA

The effects of different compatibility ratios of BA and MS (10:1, 5:1, 20:1, 20:3, 30:2, 15:2, 50:3) on IL-1β, IL-6, and iNOS mRNA expression in LPS-induced BV2 cells were determined by q-PCR (results shown in Figure 2). When The expression levels of IL-1β, IL-6, and iNOS mRNA in other proportions of compatibility were down-regulated significantly, except for the ratio of 15:1 of BA and MS it had no significant effect on IL-6.

### 2.3. Expression of p-NF-ҡB/NF-ҡB, p-AKT/AKT, p-PI3K, and TLR4 Protein

Western blot analysis was performed with different compatibility proportions (10:1, 5:1, 20:1, 20:3, 15:1, 15:2, 50:3) of BA and MS on LPS-induced BV2 cells with respect to p-NF-ҡB/NF-ҡB, p-AKT/AKT, p-PI3K, and TLR4 protein expression. The results are shown in Figure 3. Overall, the protein expression levels of p-NF-ҡB/NF-ҡB, p-AKT/AKT, and TLR4 were reduced and p-PI3K protein was up-regulated to varying degrees in BV2 cells after treatment with ratios of 5:1, 20:3, and 50:3.

### 2.4. Determination of 15 Compounds with Different Compatibility Ratios of BA and MS

The contents of 15 compounds with different compatibility ratios of BA and MS were determined by UPLC-TQ/MS (results are shown in Table 1). The contents of chemical components varied with different compatibility ratios of BA and MS; when the compatibility ratios of BA and MS was 50:3, the content of 3-acetyl-11-keto-β-boswellic acid was the highest.

### 2.5. Network Pharmacology

#### 2.5.1. Bioactive Compounds Selected

Pearson correlation coefficient values *r* > 0.65 and *r* < ‒0.65 indicate a significant positive correlation and a significant negative correlation, respectively. As shown in Figure 4, the 15 compounds were clearly distinguished in red and blue, indicating that the correlation trend between the seven effect indicators and the compounds was close to the same. Compound 1 was positively correlated with IL-1β (*r* = 0.713) and negatively correlated with IL-6 (*r* = −0.714). Compound 2 was positively correlated with p-AKT/AKT (*r* = 0.92). Compound 3 was negatively correlated with p-P56/P56 (*r* = −0.653). Compound 4 was negatively correlated with p-p56 /P56 (*r* = −0.71).

#### 2.5.2. Targets of Anti-Neuroinflammation

Screening of anti-neuroinflammatory targets for four active compounds revealed 201 targets related to neuroinflammation. Among them, 85 targets were related to neuroinflammation with Compound 1, 68 targets in neuroinflammation were associated with Compound 2, and 75 targets were related to neuroinflammation with Compound 3. There were 43 targets for neuroinflammation in Compound 4, and the common targets of four compounds against neuroinflammation were 5-lipoxygenase (ALOX5) and Androgen Receptor (AR). The protein with score > 90 was selected to determine the protein-protein interaction (PPI) with Cytoscape version 3.7.1 software. As shown in Figure 5, the size of the ellipse is represented by the degree value. After simplified analysis of topological parameters, 32 key proteins were extracted as key targets for anti-neuroinflammation, namely PI3K p110α (PIK3CA), protein kinase b 1 (AKT1), estrogen receptor 1 (ESR1), recombinant E1A binding protein P300 (EP300), mitogen-activated protein Kinase 1 (MAPK1), sphingosine 1 phosphate receptor 3 (S1PR3), mitogen-activated protein Kinase 3 (MAPK3), epidermal growth factor receptor (EGFR), Caspase-3 (CASP3), mitogen-activated protein Kinase 14 (MAPK14), coagulation factor II (F2), type-2 angiotensin II receptort (AGTR2), tumor necrosis factor (TNF), protein kinase C alpha type (PRKCA), type-1 angiotensin II receptort (AGTR1), mitogen-activated protein Kinase 8 (MAPK8), recombinant nuclear receptor subfamily 3, group c, member 1 (NR3C1), protein kinase C delta type (PRKCD), cannabinoid receptor 1 (CNR1), endothelin receptor type B (EDNRB), glutamate metabotropic receptor 1 (GRM1), protein kinase C zeta type (PRKCZ), tyrosine-protein kinase JAK2 (JAK2), 5-hydroxytryptamine receptor 2A (HTR2A), D(2) dopamine receptor (DRD2), tyrosine-protein kinase JAK1 (JAK1), C3a anaphylatoxin chemotactic receptor, C3AR, C3a-R (C3AR1), tyrosine-protein kinase JAK3 (JAK3), substance-P receptor (TACR1), tyrosine-protein kinase (TYK2), cannabinoid receptor 2 (CNR2), and fMet-leu-phe receptor (FPR1).

#### 2.5.3. Kyoto Encyclopedia of Genes and Genomes (KEGG) Pathway 

On analysis of 32 selected key targets by the DAVID database (DAVID.ncifcrf.gov) according to *p*-value and false discovery rate (FDR), 31 signaling pathways were screened as shown as Table 2. The “bioactive component–target–pathway” network diagram is shown in Figure 6. The key targets included MAPK3, MAPK1, PIK3CA, AKT1, MAPK14, MAPK8, PRKCA, EGFR, and JAK1 (degree ≥ 10) and the key pathways included the neuroactive ligand–receptor interaction, influenza A, pathways in cancer, and Rap1 signaling pathway.

#### 2.5.4. Biological Process Enrichment

Gene Ontology (GO) biological process enrichment analysis was performed on 32 selected key targets using the DAVID database (DAVID.ncifcrf.gov), as shown in Figure 7. The GO enrichment analysis found a relationship with protein phosphorylation, intracellular signal transduction, plasma membrane components, and ATP binding.

## 3. Discussion

Frankincense and myrrha compatibility showed synergistic effects with the commonly used ratios of frankincense–myrrha (FM) of 1:1, 1:2, 2:1, 2:3, 3:2, 3:4, 5:3, as shown by analysis of all the formulae containing FM in the database of prescriptions of Nanjing University of Chinese Medicine. According to the proportions of common clinical compatibility of frankincense and myrrh, the BA and MS compatibility ratios (the effective part of frankincense myrrh) of 10:1, 5:1, 20:1, 20:3, 15:1, 15:2, and 50:3 were used in this study. When the compatibility ratios of BA and MS were different, there were some differences in the contents of chemical ingredients. Therefore, there are relationships between the contents of ingredients and the effective indexes on BV2 cells, and Pearson correlation heat map analysis can intuitively be used to establish these intrinsic relationships. Through the existing open databases (TCSMP, SwissTargetPrediction, Gene cards, STRING, DAVID), the target information of Pearson-related (r > 0.65, r < ‒0.65) compounds in BA and MS compatibility ratios was analyzed. PPI interaction maps were constructed and topological analysis of biological information were performed to obtain a compound–target–pathway relationship diagram.

The Pearson correlation coefficient is a statistic that reflects the degree of linear correlation between two variables [21]. The correlation coefficient is represented by *r*, and the larger the absolute value of *r*, the stronger the correlation. Therefore, the Pearson correlation coefficient analysis method can be used to determine the correlation between the content of multiple compounds in Chinese medicines and multiple effect indicators, and the important efficacy indicators and key compounds corresponding to the “effect” can be selected according to the correlation intensity [22]. Based on the above ideas, the biological characteristics of LPS-induced BV2 cells were adopted, and UPLC-TQ/MS was combined to determine the content of 15 compounds in different proportions of BA and MS. Moreover, four key compounds were screened using Pearson correlation analysis.

The four compounds have been confirmed to have obvious bioactive components. Compound 1 was isolated and identified for the first time by Su, and it was found to have significant activity in inhibiting aromatase in human umbilical vein endothelial cells (ECV304) under an H_2_O_2_ environment [23]. Compound 2 could induce inflammation through induction of granulation in rat neutrophils [24], and was also found reduce the expression of IL-4 [25]. Compound 3 nanoparticle preparation has significant anti-inflammatory activity [26], inhibits the NO produced by LPS-induced leukemia cellsin mouse macrophage (RAW264.7) cells, and exhibits certain anti-inflammatory activity [27]. Recent studies have widely focused on Compound 4 because of its anti-inflammatory effects, as it enhances IL-1α expression by activating matrix metalloproteinase-9 [28] and has also been reported to promote the regeneration of sciatica, which may be related to the phosphorylation of extracellular regulated protein kinases (ERK) protein [29].

Through KEGG signaling pathway analysis, it was found that seven targets were related to the Toll-like receptor signaling pathway, namely AKT1, MAPK1, TNF, MAPK14, MAPK3, PIK3CA, and MAPK8, indicating that the ratio of the BA and MS may be adjust the Toll-like receptor signaling pathway to exert its anti-neuroinflammatory effects. Among them, AKT1 is one of the three subtypes of AKT [30], and AKT1 can control the response of macrophages to lipopolysaccharide by regulating miRNA [31]. PIK3CA encodes PI3Kp110 α, and mutations can lead to the activation of PI3Ks, thereby reducing apoptosis. TLR4 belongs to the Toll-like receptor familyand is well known as the unique receptor for LPS [32]. The role of TLR4 in LPS-mediated signal transduction has been widely studied [33,34]. Microglia express TLR4 under the condition of LPS stimulation [35], and then activate NF-ҡB through signal transduction; AKT and PI3K proteins involved in apoptosis are also activated [35]. Promoting the release of IL-1β, IL-6, iNOS and other inflammatory cytokines thereby leads to inflammatory damage. The in vitro results showed that the IL-1β, IL-6, and iNOS mRNA expression was reduced and the TLR4/PI3K/AKT/NF-ҡB signaling pathways were down-regulated with different ratios of BA and MS, consistent with network pharmacology analysis. It was further confirmed that the neuroprotective effects and mechanisms of different ratios of BA and MS to LPS-induced BV2 cells may be related to the down-regulation of the Toll-like receptor signaling pathway.

## 4. Materials and Methods

### 4.1. Experimental Materials

A Waters ACQΜITY UPLC System (including quaternary pump solvent system, online degasser and autosampler; Waters, USA), Xevo Mass Spectrometer (Waters), MassLynx 4.1 Mass Spectrometry Workstation Software (Waters), and Anke LXJ-IIB type and TDL-240B centrifuges were purchased from the Shanghai Anting Scientific Instrument Factory (Shanghai, China). An EPED Ultrapure Water System (Nanjing Yipudayi Technology Development Co., Ltd. Nanjing, China), BT125 Electronic Balance (Sedolis Scientific Instruments Co., Ltd.), Dai Sheng High-Speed Universal Pulverizer (Yongkang City Jiushun Ying Trading Co., Ltd. Yongkang, China), Forma Series II Water Jet Type CO2 Incubator (Thermo Fisher Scientific, USA), 1300 Series A2 Ultra-Clean Workbench (Thermo Fisher Scientific), PrimoStar Inverted Microscope (ZEISS), TOMY SX-500 Autoclave (Nanjing Jitian Biotechnology Co., Ltd. Nanjing, China), Microplate Constant Temperature Oscillator (Thermo Fisher Scientific), EnSpire® Multimode Plate Reader (PerkinElmer, USA), high-speed centrifuge (Allegra X-12R Centrifuge and Microfuge 22R Centrifuge, Backman), MINI-4K Micro Centrifuge (Hangzhou Miou Instrument Co., Ltd. Hangzhou, China), BWS-10 Constant Temperature Water Bath (Shanghai Yiheng Scientific Instrument Co., Ltd. Shanghai, China), Ultrapure Water Preparation Instrument (Millipore, USA), Pipetting Gun (Eppendorf), WGL-230B Electric Blast Drying Oven (Tianjin Taisite Instrument Co., Ltd. Tianjin, China), X025-12DT Ultrasonic Cleaner (Nanjing Xianou Instrument Co., Ltd. Nanjing, China), SIM-F140AY65-PC Ice Machine (Japan Matsushita Electric Industrial Co., Ltd.), Biorad Mini-PROTEAN Tetra Electrophoresis System (BioRad, USA), Rocker Type Bleaching Shaker (Haimen Qilin Bell Instrument Manufacturing Co., Ltd.), Metal Bath (Stuart, UK), ChemiDoc Imaging System (BioRad), Real-Time Quantitative PCR (ABI 7500), and Ultra-Micro-Ultraviolet Spectrophotometer (DS-11) were also used.

Frankincense (lot 171010) and myrrh (lot 171108) were produced in Guangxi, and were purchased from the Suzhou Tianling Chinese Herbal Medicine Co., Ltd. (Suzhou, China). The processing method used vinegar. They were identified by Professor Duan Jin-ao of Nanjing University of Traditional Chinese Medicine. Frankincense is a resin that exudes from *Bowiellia carteri* Birdw, and myrrha is a dried resin of *Commiphora myrrha* Engl.

The frankincense standards included 3-oxotirucall-8,24-dien-21-oic acid, batch number 18032302, purity ≥98%; 3*α*-acetoxy-tirucall-7,24-dien-21-oic acid, batch number 15698198, purity ≥98%; acetyl 11*α*-methoxy-*β*-boswellic acid, batch number 15687198, purity ≥98%; 3*α*-hydroxy tirucall-7,24-dien-21-oic acid, batch number 15695198, purity ≥98%; 11-keto-boswellic acid, batch number 18032113, purity ≥98%; and 3-acetyl-11-keto-*β*-boswellic acid, batch number 18032706, purity ≥98%. All of the above were purchased from Nanjing Liangwei Biotechnology Co., Ltd. (Nanjing, China). 3-hydroxytirucall-8,24-dien-21-oic acid, batch number 20150520, purity ≥98%; 3-*O*-acetyl-*α*-boswellic acid, batch number 20150515, purity ≥98%; 3*α*-acetyloxylanosta-8,24-dien-21-oic acid, batch number 20150515, purity ≥98%; 3*β*-acetoxy-5*α*-lanosta-8,24-dien-21-oic acid, batch number, 20150527, purity ≥98%; 3-acetyloxy-tirucall-8,24-dien-21-oic acid (self-made in the laboratory), purity ≥95%; *α*-boswellic acid, batch number 20150522, purity ≥98%; and *β*-boswellic acid, batch number 20150504, purity ≥98% were all purchased from Baoji Chenguang Biotechnology Co., Ltd. (Baoji, China).

The standard products of myrrh were 2-methoxy-8,12-epoxygermacra-1(10),7,11-trien-6-one, self-made in the laboratory (purity ≥95%); and 2-methoxy-5-acetoxy- fruranogermacr-1(10)-en-6–one, self-made in the laboratory (purity ≥95%).

LPS (CST, Lot No. 14011 S), was purchased from Nanjing Formes Biotechnology Co. Ltd., China, Nanjing). MEM culture medium, antibiotics (penicillin 10,000 μg·mL^−1^ and streptostreptin 10,000 μg·mL^−1^, P/S), fetal bovine serum, and sodium pyruvate were all purchased from Gibco, USA. Dimethyl sulfoxide (DMSO), thiaxolyl blue tetraxoliu, 3-(4,5-Dimethylthiazol-2-yl)-2,5-diphenyltetrazolium bromide (MTT), and TRIZOL reagent were purchased from Sigma, USA. The cDNA reverse transcription kit and qPCR kit were purchased from Beijing Quanxing Biotechnology Co., LTD. (Beijing, China). The PAGE Gel Rapid Preparation Kit (10%) was purchased from Shanghai Yazyme Biotechnology Co., Ltd. (Shanghai, China). Tris base (Affymetrix, USA), glycine, and BCA kit were purchased from Beijing Suo Laibao Biotechnology Co., Ltd. (Beijing, China). Tween 20 (Sigma, USA) and bovine serum albumin (BSA) were purchased from MP Biomedicals. RIPA lysate and T-BST were purchased from Wuhan Google Biotechnology Co., Ltd., (Wuhan, China). ECL chemiluminescence was purchased from Shanghai Tianneng Technology Co., Ltd., (Shanghai, China). Anti-rabbit anti-mouse TLR4 (Proteintech), NF-ҡB-P65, p-NF-ҡB-P65, p-AKT, AKT, p-PI3K, β-actin, secondary antibody horseradish peroxidase (HRP), and sheep anti-rabbit (CST, USA) were also used. Chloroform, isopropanol, methanol, and ethanol were all analytically pure. Acetonitrile and methanol were chromatographically pure.

### 4.2. Methods

#### 4.2.1. Preparation of Drug Solution

BA preparation: Frankincense was extracted by 95% ethanol four times, and the time of extraction was 1 h each time. Then the supernatants were mixed and the solvent evaporated to obtain dry extracts. BA was then obtained by alkali-soluble acid precipitation.

MS preparation: Myrrha was extracted by 92% ethanol twice; the extraction time was 2.5 h each time. The supernatants were mixed and evaporated the solvent to obtain dry extracts. Then the ethyl acetate extract was purified by silica gel column chromatography to obtain MS.

Based on the seven compatibility ratios of FM (1:1, 2:1, 1:2, 2:3, 3:2, 3:4 and 5:3.) in the clinic, with BA and MS extraction efficiency conversion values of 20% and 2%, respectively, BA and MS compatibility ratios of 10:1, 20:1, 5:1, 20:3, 15:1, 15:2 and 50:3 were calculated. BA and MS were together dissolved in 1% dimethyl sulfoxide (DMSO) to prepare 100 mg/mL of the 10:1, 5:1, 20:1, 20:3, 15:1, 15:2 and 50:3 technical liquid concentrates for storage.

#### 4.2.2. Cell Culture

BV2 cells were purchased from the American Type Culture Collection (VA, USA), and were cultured in MEM medium containing 10% FBS, 1% P/S and 1% sodium pyruvate, and placed in an incubator containing 5% CO_2_ at a constant temperature at 37 °C. The cells were used for experiments when they grew to the logarithmic growth stage.

#### 4.2.3. MTT

BV2 cell suspension was inoculated in 96-well plates with 5×10^4^ cells /mL and 100 μL/well; each of the groups was evaluated in triplicate. The groups included the control and groups with different ratios of BA and MS (10:1, 5:1, 20:1, 20:3, 15:1, 15:2, 50:3), and different concentrations (0.75, 1.5, 3, 6.25, 12.5, 25, 50 μg·mL^−1^). After 24 h, 10 μL/well MTT solution were added so that finally the MTT concentration of each well was at 0.5 μg·mL^−1^, and the volume was 100 μL. After 4 h of incubation in the dark, the supernatant was removed, and 150 μL DMSO was added to the wells. The next step was incubation for 30 min in the dark on a shaker to measure the absorbance at 570 nm.

#### 4.2.4. q-PCR

The BV2 microglia suspension was inoculated in six-well plates at 10^5^/ well, with 2 mL per well, and the supernatants were discarded 24 h later. The cells were divided into the control group, model group (LPS 1 μg·mL^−1^), and therapy group (10 μg·mL^−1^, 3 μg·mL^−1^, 1 μg·mL^−1^) + LPS 1 μg·mL^−1^ for 24 h, and the cells were collected overnight at ‒80 °C. RNA was extracted, reverse transcribed into cDNA, and then amplified to detect the expression of IL-1β, IL-6, and iNOS mRNA, repeated three times. Primer sequences of each mRNA are as follows:

IL-6: F: 5′- CTG CAA GAG ACT TCC ATC CAG-3′

  R: 5′-AGT GGT ATA GAC AGG TCT GTT GG-3′

IL-1β: F: 5′- TGCCACCTTTTGACAGTGATG-3′

  R: 5′-TGATACTGCCTGCCTGAAGC-3′

iNOS: F: 5′-GTT CTC AGC CCA ACA ATA CAA GA-3′

  R: 5′-GTG GAC GGG TCG ATG TCA C-3′

GAPDH: F: 5’-ATC ATC TCC GCC CCT TCT G-3′

  R: 5’-GTG ATG GCA TGG ACT GTG G-3′

#### 4.2.5. Western Blot

The BV2 cell suspension was inoculated into a sis-well plate at 10^4^/well, 2 mL per well. After 24 h, the supernatants were discarded and the cells were divided into a control group and model group (LPS 1 μg·mL^−1^). After 48 h of administration in the drug-administered group (3 μg·mL^−1^ + 1 μg·mL^−1^ LPS), the cells were collected and incubated at ‒80 °C overnight. The protein of cells was extracted and the protein concentration was determined by BCA assay. The protein was inactivated at 95 °C for 15 min. Then treatment was performed in succession with SDS-PAGE gel electrophoresis, wet transposition, 5% BSA blocking for 2 h, incubation at 4 °C primary antibody overnight, room temperature equilibrium for 1 h, and secondary antibody incubation for 1 h. After a 2-min reaction with chemiluminescence detection reagents, the membrane was sensitized and developed with X film in the darkroom.

#### 4.2.6. Chromatographic Conditions

##### 4.2.6.1. Analysis Condition of UPLC-TQ/MS

Chromatographic conditions: ACQUITY UPLC BEH C18 column (2.1 mm × 100 mm, 1.7 μm); Column temperature: 30 °C; flow rate: 0.4 mL·min^−1^; injection volume: 2 μL; mobile phase was composed of (A) formic acid solution (0.1%) and (B) acetonitrile using a gradient elution of 90% B at 0–1 min, 90–91% B at 1–3 min, 91–92% B at 3–9 min, 92–95% B at 9–12 min, 95–90% B at 12–14 min, 95–90% B at 14–15 min.

MS conditions: Electrospray positive and negative ion sources (ESI^+^/ESI^−^), multi-reaction detection (MRM method), helium flow rate: 35 arb, auxiliary gas flow rate: 15 arb, capillary temperature: 275 °C, spray voltage: 3.5 kV, tube lens voltage level: 65%, capillary voltage: 35 V, scanning range: *m/z* 100~1000. Mass resolution: 60,000. The secondary mass spectrometry analysis used cleavage mode: high energy-induced dissociation (HCD); isolation width: 2.0 Da; normalized collision energy: 35 V; activation time: 30 ms; collision gas: high purity nitrogen. The main parameters of mass spectrometry are shown in Appendix A, and the ion flow chromatogram of the mixed reference and sample are shown in Appendix A.

##### 4.2.6.2. Solution Preparation

After the frankincense and myrrha medicinal materials were passed through a 40-mesh sieve, the medicinal material powders were accurately weighed to prepare the extractions of BA and MS respectively, according to the determined extraction and purification method. Then, the BA and MS extractions were converted into 10:1, 5:1, 20:1, 20:3, 15:1, 15:2, 50:3 compatibility ratios, and were dissolved with the proper amount of methanol, Next, the sample solution above were centrifuged at 13,000 r·min^−1^ for 10 min twice, and the upper liquid was taken to filter by using the microporous membrane of 0.22 μm. Finally, the sample was diluted 20 times for analysis.

##### 4.2.6.3. Mixed Reference Solution

The reference substances of frankincense and myrrha were weighed accurately and were dissolved in methanol to obtain stock solutions of 1 mg·mL^−1^ concentration. Then, the mixed reference solution for analysis was obtained by diluting and mixing the standard stock solutions (1 mg·mL^−1^) appropriately. It contained 2-methoxy-8,12-epoxygermacra-1(10),7,11-trien-6-one 45.6 μg·mL^−1^, 2-methoxy-5 -acetoxy-fruranogermacr-1(10)-en-6–one 45.2 μg·mL^−1^, 3-oxotirucall-8,24-dien -21-oic acid 52.4 μg·mL^−1^, 3*α*-acetoxy-tirucall-7,24-dien-21-oic acid 34.0 μg·mL^−1^, 3-hydroxytirucall-8,24-dien-21-oic acid 40.8 μg·mL^−1^, acetyl 11*α*-methoxy -*β*-boswellic acid 36.5 μg·mL^−1^, 3*α*-hydroxy tirucall-7,24-dien-21-oic acid 38.8 μg·mL^−1^, 11-keto-boswellic acid 41.6 μg·mL^−1^, 3-*O*-acetyl-*α*-boswellic acid 53.2 μg·mL^−1^, 3*α*-acetyloxylanosta-8,24-dien-21-oic acid 44.0 μg·mL^−1^, 3*β*-acetoxy-5*α* -lanosta-8,24-dien-21-oic acid 25.9 μg·mL^−1^, 3-acetyl-11-keto-*β*-boswellic acid 47.2 μg·mL^−1^, 3-acetyloxy-tirucall-8,24-dien-21-oic acid 34.0 μg·mL^−1^, *α*-boswellic acid 41.2 μg·mL^−1^, and *β*-boswellic acid 50.4 μg·mL^−1^.

##### 4.2.6.4. Methodological Investigation

To investigate the linear relationship, the mixed reference solution was diluted 2, 4, 8, 16, 32, 64, 128, 256, 512, 1024, and 2048 times, filtered using a 0.22-μm microporous filter membrane, and analyzed in turn. Taking the peak area as a longitudinal coordinate and the concentration of reference substance as transverse coordinate, the standard curve was drawn and the regression equation was calculated. The results are shown in Appendix A.

To determine the precision, the mixed reference solution was taken and was injected 6 times continuously. Then the chromatographic peaks of each standard substance were recorded. The relative standard deviation (RSD) of each major peak area were less than 5.9%, indicating that the instrument precision was good.

Repeatability measurements were performed using six samples of sample solution with BA and MS of 10:1 in parallel, and the sample was injected successively for determination. The peak area of each sample to be measured was recorded, and the content was calculated. The results showed that the RSD of each content was less than 5.1%, indicating that the method had good reproducibility.

In the stability experiment, sample solutions were taken and injected for determination at 0, 2, 4, 6, 8, and 10 h, and chromatographic peaks of all standard substances were recorded. The RSD of each content was less than 5.8%, indicating that the test solution was chemically stable within 10 h.

The average recovery was established by extracting six samples with a 10:1 ratio of BA and MS according to the sample preparation method, and the reference samples with the same amount of compounds in the samples were added with precision. The average recovery rate of each index component was 84.3~103.9%, and the RSD was 1.6~4.9%.

#### 4.2.7. Mechanism Study of Network Pharmacology

##### 4.2.7.1. Screening Effective Components

We searched the Traditional Chinese Medicine Systems Pharmacology (TCMSP) Database and Analysis Platform and used UPLC-TQ/MS to determine the content of related compounds reported in the literature in the BA and MS, combined with the biological effect index of BV2 cells to Pearson-related thermographs by using Spss24.0 and GraphPad Prism7, and then chose the compounds with a Pearson correlation coefficients *r* > 0.65 and *r* <-0.65 as the key active ingredients.

##### 4.2.7.2. Target Screening

The structures of the four active compounds were input into SwissTargetPrediction (www.swisstargetprediction.ch), and the targets of probability > 0 were selected as the targets of the compound. A total of 325 related targets were obtained. The key word “nerve inflammation” was inputted in the Gene card database and 6866 target genes related to neuroinflammation were obtained. On mapping 6686 neuroinflammation target genes and 325 drug target genes we obtained 201 neuroinflammation-related targets. Then, we inputted the 201 nerve inflammation-related targets into the STRING (string-db.org) database, selected the species as “*Homo sapiens*”, picked the protein with score of >90, and plotted the PPI protein interaction map with Cytoscape version 3.7.1 software. Through the Network Analyzer tool, the topological parameters of the target in the network were analyzed, and the degree were greater than the median of two times. The betweenness centrality (BC) and the closeness centrality (CC) were greater than the median and were the screening conditions to simplify the PPI.

##### 4.2.7.3. KEGG Pathway Enrichment Analysis

The 32 key targets screened under “Section 4.2.7.2” were analyzed by the DAVID database (david.ncifcrf.gov). setting the species to “*Homo sapiens*”. The KEGG pathway was enriched by selecting *p* ≤ 0.01 and FDR ≤ 0.01. As shown in Table 2, a total of 31 signaling pathways were screened. The selected KEGG pathway and active compounds were used to construct the “component-target-pathway” network diagram with Cytoscape_3.7.1 software. The key targets and pathways are predicted by values and mediums.

##### 4.2.7.4. Biological Process Enrichment Analysis

The biological pathway analysis was carried out using the DAVID database (david.ncifcrf.gov) to set the species to be “*Homo sapiens*”, with the selection of *p* ≤ 0.01 and FDR ≤ 0.01 to carry on the enrichment analysis of the GO biological process, wherein the enrichment analysis of the GO biological process includes the biological process, molecular function, and cellular component.

### 4.3. Statistic Analysis

SPSS24.0 software was used for mathematical statistical analysis of the data. The data are expressed as mean ± SD. One-way analysis of variance (ANOVA) was used for comparison between the groups. Bilateral *p*-values less than 0.05 were considered to be statistically significant. The model group and control group were considered to indicate statistical significance at ^#^*p* < 0.05, ^##^*p* < 0.01, ^###^*p* < 0.001. Values of **p* < 0.05, ***p* < 0.01, ****p* < 0.001 indicate statistical significance between the treatment group and the model group.

## 5. Conclusions

In this article, we revealed the anti-neuroinflammatory effects and mechanisms of boswellic acid (BA) and myrrha sesquiterpenes (MS) with different proportions compatibility based on an LPS-induced BV2 cell model. Among different compatibility ratios of BA and MS, it was found that ratios of 10:1, 5:1, and 20:1 exhibited better inhibitory effects. It was found that 2-methoxy-5-acetoxy-fruranogermacr-1(10)-en-6-one, 3*α*-acetyloxylanosta-8,24-dien-21-oic acid, 11-keto-boswellic acid, and 3-acetyl-11-keto-*β*-boswellic acid were the main bioactive ingredients by Pearson correlation analysis. Thirty-two key targets and 31 related pathways were obtained by network pharmacology, and the anti-neuroinflammatory effects of these four bioactive compounds were found to inhibit IL-1β, IL-6, and iNOS mRNA expression, down-regulate the p-NF-ҡB/NF-ҡB, p-AKT/AKT, TLR4 protein expression level, and increase the p-PI3K protein expression level. More experimental data in future studies would further verify the effects and mechanisms.

## Figures and Tables

**Figure 1 molecules-24-03946-f001:**
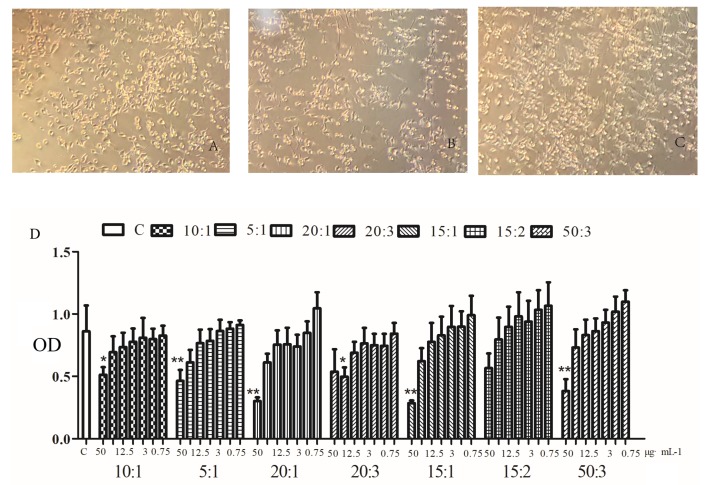
Cell morphology of Normal control group (**A**), Model group (**B**) and Administration group with boswellic acid (BA): myrrha sesquiterpenes (MS) (10:1) drug (**C**) (×200). Effects of different ratios of boswellic acid (BA) and myrrha sesquiterpenes (MS) on cytotoxicity (**D**).

**Figure 2 molecules-24-03946-f002:**
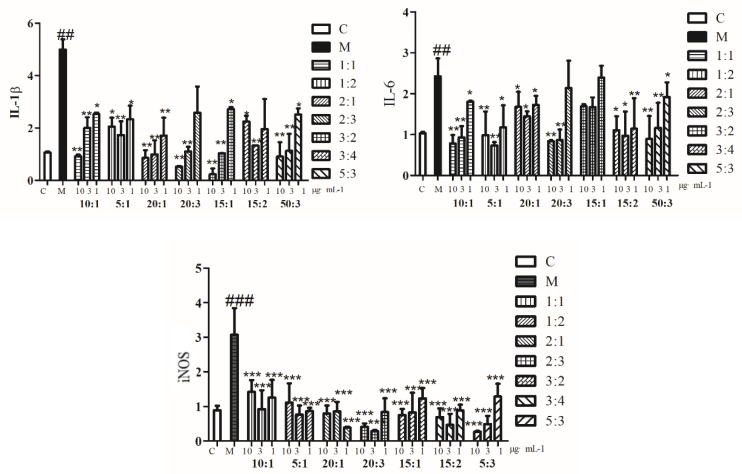
The expression of IL-1β, IL-6, and iNOS mRNA in BV2 cells induced by lipopolysaccharide (LPS) was significantly increased in the model group (#*p* < 0.05, ##*p* < 0.01, ###*p* < 0.001). Compared with the model group, the expression level of IL-1β, IL-6, and iNOS mRNA in the administration group was significantly lower than that in the model group (**p* < 0.05, ***p* < 0.01, ****p* < 0.001).

**Figure 3 molecules-24-03946-f003:**
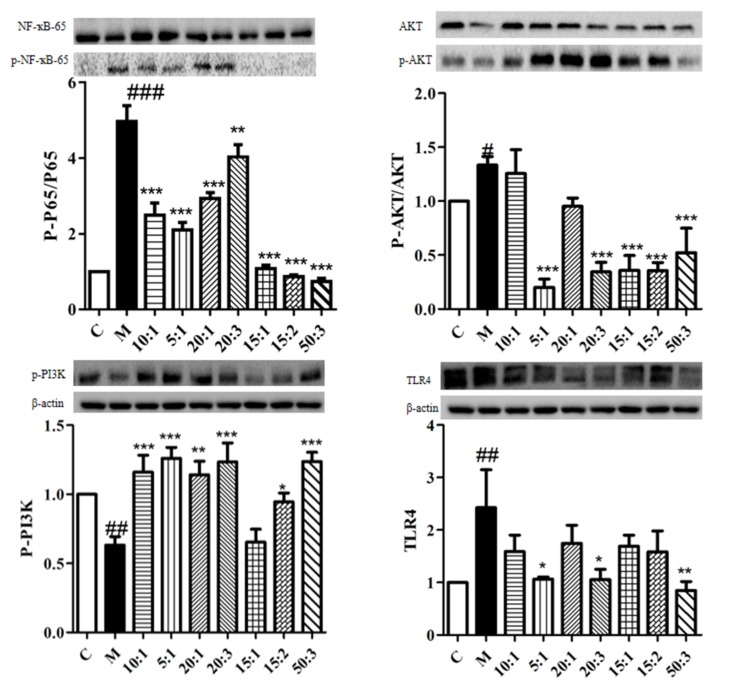
Compared with the control group, the expression of p-NF-ҡB/NF-ҡB, p-AKT/AKT, and TLR4 protein was significantly increased, and the expression of p-PI3K protein was significantly decreased (#*p* < 0.05, ##*p* < 0.01, ###*p* < 0.001). Compared with the model group, the expression of p-NF-ҡB/NF-ҡB, p-AKT/AKT, p-PI3K, and TLR4 protein was significantly decreased, and the expression of p-PI3K protein was significantly increased (**p* < 0.05, ***p* < 0.01, ****p* < 0.001).

**Figure 4 molecules-24-03946-f004:**
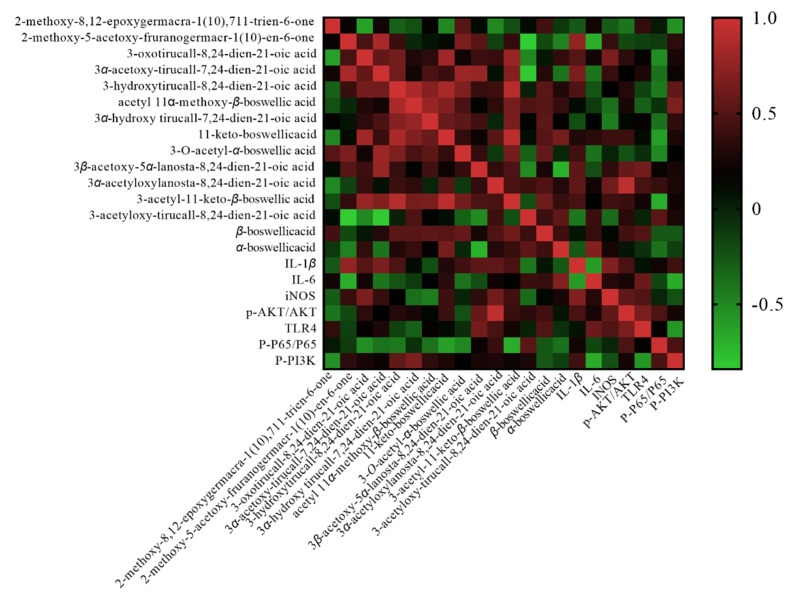
Heat map of biological effect indexes of BV2 cells, using the main chemical components in BA and MS.

**Figure 5 molecules-24-03946-f005:**
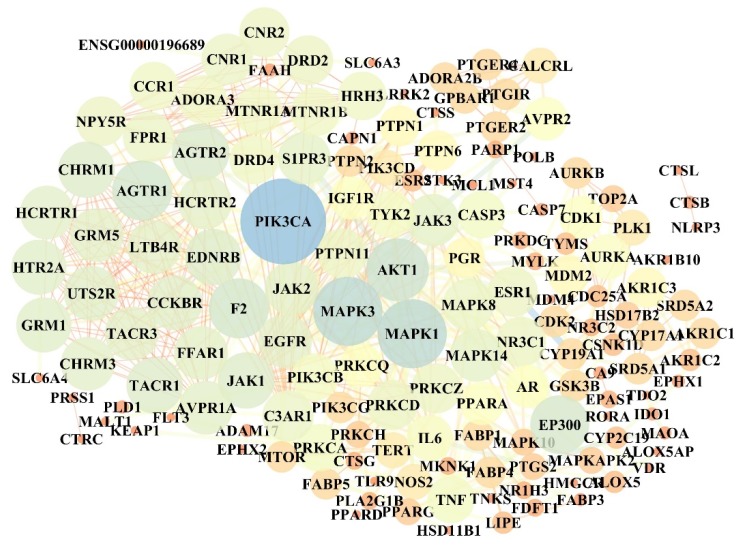
protein-protein interaction (PPI) diagram of target protein.

**Figure 6 molecules-24-03946-f006:**
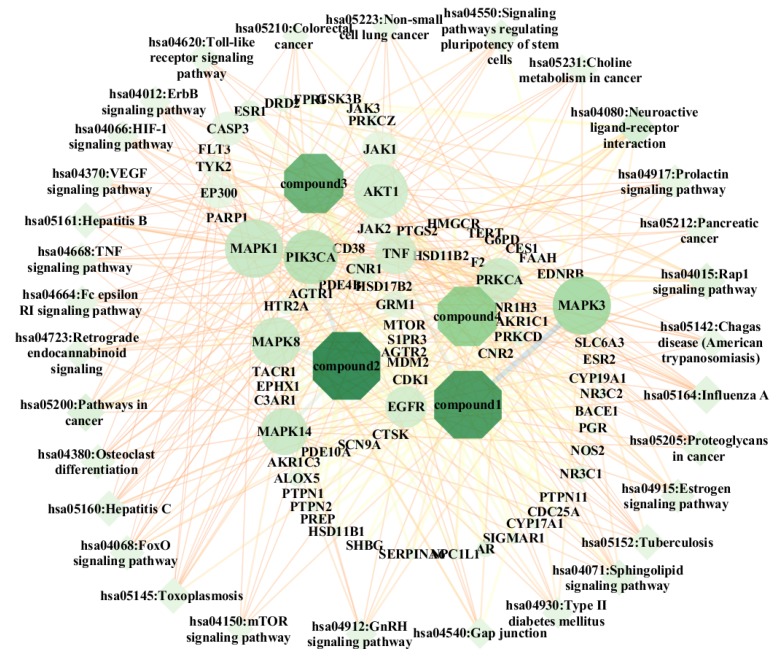
Bioactive component-target-pathway (octagons represent bioactive components, ellipses represent targets, diamonds represent pathways and nodes represent compounds, targets, and pathways; edges represent the interaction of compound-targets, targe- pathways).

**Figure 7 molecules-24-03946-f007:**
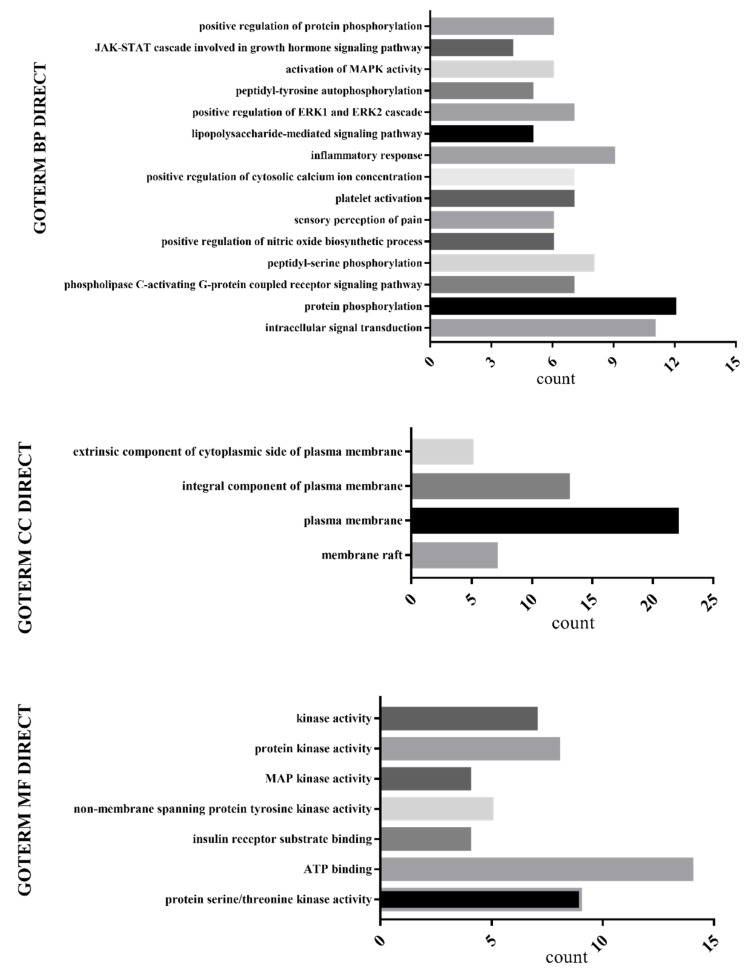
GO biological process enrichment analysis.

**Table 1 molecules-24-03946-t001:** Content of compounds with different proportions of BA and MS.

Compound	Compatibility Ratio (%)						
	10:1	5:1	20:1	20:3	15:1	15:2	50:3
2-methoxy-8,12-epoxygermacra-1(10),7,11-trien-6-one	0.30	0.32	0.15	2.58	2.48	3.13	0.30
2-methoxy-5-acetoxy-fruranogermacr-1(10)-en-6–one	3.85	5.91	1.41	2.46	2.32	4.91	2.55
3-oxotirucall-8,24-dien-21-oic acid	2.76	2.95	2.64	1.27	2.39	2.47	2.87
3*α*-acetoxy-tirucall-7,24-dien-21-oic acid	0.87	0.87	0.69	0.64	0.69	1.02	0.77
3-hydroxytirucall-8,24-dien-21-oic acid	1.00	0.91	0.87	0.83	0.81	0.98	1.08
acetyl 11*α*-methoxy-*β*-boswellic acid	1.31	1.07	1.23	1.27	0.90	1.28	1.78
3*α*-hydroxy tirucall-7,24-dien-21-oic acid	1.43	1.33	1.77	1.45	1.28	2.10	2.10
11-keto-boswellicacid	1.33	1.20	1.25	1.02	1.22	1.23	1.47
3-*O*-acetyl-α-boswellic acid	3.17	2.67	0.36	2.50	2.22	5.10	3.68
3*α*-acetyloxylanosta-8,24-dien-21-oic acid	0.49	0.41	0.47	0.34	0.31	0.54	0.32
3*β*-acetoxy-5*α*-lanosta-8,24-dien-21-oic acid	3.40	1.60	2.41	1.81	1.89	1.51	2.13
3-acetyl-11-keto-*β*-boswellic acid	7.71	6.44	6.44	5.14	6.39	7.57	8.06
3-acetyloxy-tirucall-8,24-dien-21-oic acid	4.10	0.20	5.27	8.04	4.31	0.45	6.32
*α*-boswellicacid	6.30	2.74	4.50	4.98	5.60	6.20	6.06
*β*-boswellicacid	1.73	1.45	1.72	1.48	2.64	1.11	2.82

**Table 2 molecules-24-03946-t002:** KEGG signaling pathway in **the database for annotation, visualization and integrated discovery** (DAVID).

ID	KEGG Signaling Pathway	Count	*p*-Value	FDR)
hsa04080	Neuroactive ligand–receptor interaction	14	6E-11	7.15E-08
hsa05164	Influenza A	12	1.1E-10	1.29E-07
hsa05145	Toxoplasmosis	10	7.4E-10	8.85E-07
hsa04071	Sphingolipid signaling pathway	10	1.6E-09	1.94E-06
hsa05160	Hepatitis C	10	0.000000004	4.85E-06
hsa05161	Hepatitis B	10	8.7E-09	1.04E-05
hsa04015	Rap1 signaling pathway	11	0.000000014	1.72E-05
hsa04664	Fc epsilon RI signaling pathway	8	0.000000015	1.76E-05
hsa04917	Prolactin signaling pathway	8	0.00000002	2.39E-05
hsa05152	Tuberculosis	10	0.00000005	5.99E-05
hsa04930	Type II diabetes mellitus	7	0.000000054	6.47E-05
hsa04380	Osteoclast differentiation	9	0.000000076	9.13E-05
hsa04068	FoxO signaling pathway	9	0.000000091	1.09E-04
hsa05205	Proteoglycans in cancer	10	0.00000014	1.72E-04
hsa04915	Estrogen signaling pathway	8	0.0000002	2.45E-04
hsa05212	Pathways in cancer	7	0.00000034	4.13E-04
hsa04668	TNF signaling pathway	8	0.00000035	4.19E-04
hsa05200	Pathways in cancer	12	0.00000055	6.62E-04
hsa04012	ErbB signaling pathway	7	0.000002	0.002354376
hsa04540	Gap junction	7	0.0000021	0.00251873
hsa04550	Signaling pathways regulating pluripotency of stem cells	8	0.0000022	0.002606689
hsa04912	GnRH signaling pathway	7	0.0000026	0.003068946
hsa04066	HIF-1 signaling pathway	7	0.0000035	0.004201762
hsa05223	Non-small cell lung cancer	6	0.0000043	0.005164264
hsa05231	Choline metabolism in cancer	7	0.0000047	0.005654591
hsa04723	Retrograde endocannabinoid signaling	7	0.0000047	0.005654591
hsa04150	mTOR signaling pathway	6	0.0000051	0.006155785
hsa05142	Chagas disease (American trypanosomiasis)	7	0.0000056	0.006706702
hsa04620	Toll-like receptor signaling pathway	7	0.0000063	0.007492714
hsa04370	VEGF signaling pathway	6	0.0000066	0.007916775
hsa05210	Colorectal cancer	6	0.0000072	0.008584012

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
