# Peer review of "Protective Effect and Mechanism of Boswellic Acid and Myrrha Sesquiterpenes with Different Proportions of Compatibility on Neuroinflammation by LPS-Induced BV2 Cells Combined with Network Pharmacology"

_molecules, 2019, doi:10.3390/molecules24213946_

Round 1
Reviewer 1 Report
In this study, Miao et al. investigated on the effects of anti-inflammatory effects and mechanisms of boswellic acid (BA) and myrrha sesquiterpenes (MS) at different ratio in BV2 cells and its potential network. The ratios of BA and MS 10:1, 5:1 and 20:1 showed higher efficacy on the expression of IL-1β, IL-6 and iNOS mRNA, and on the p-NF-ҡB/NF-ҡB, p-30, p-AKT/AKT, TLR4, and the p-PI3K. In addition, pearson correlation analysis between the effects and 15 bioactive substances predicted that the compounds of 2-methoxy-5-acetoxy -fruranogermacr-1(10)-en-6-one (compound 1), 3α-acetyloxylanosta-8,24-dien-21-oic acid (compound 2), 11-keto-boswellic acid (compound 3), 3-acetyl-11-keto-β -boswellic acid (compound 4) were important contributions of bioactive components for treatment on neuroinflammation. These four active compounds act on neuroinflammatory related targets and involves 32 signaling pathways based on network pharmacology analysis. This study seems potentially interesting, however, some points should be revised.
Points
The resolution of Fig. 1 is low. It should be replaced with another photo with higher resolution. Significance should marked into the graph of Fig.1 2. The resolution of western bands of Fig. 3 should be improved. 3.English should be revised by native speaker(s).
Author Response
Thank you for your kind suggestion. We have improved the resolution of Fig1 and Fig3, and the graph in Fig1(2) has been marked with significance, and corrected the language with the help of native English speakers.
Reviewer 2 Report
The manuscript reported the anti-inflammatory effects of the compounds of 2-methoxy-5-acetoxy -fruranogermacr-1(10)-en-6-one (compound 1), 3α-acetyloxylanosta-8,24-dien-21-oic acid (compound 2), 11-keto-boswellic acid (compound 3), 3-acetyl-11-keto-β -boswellic acid (compound 4) in cells and were reported to be important bioactive components for treatment on neuroinflammation. However, the studies are not complete. The mechanistic actions of a mixture of compounds cannot be clearly characterized. Although the neuroinflammatory related targets and the associated signaling pathways based on network pharmacology analysis were described but it is far from clear. The conclusion needs to be substantiated with more supportive data in order to provide a basis for further research and the development of new antineuroinflammatory drug. Finally, more updated references on the subject matters should be cited.
Author Response
Thanks for the reviewer’s professional comments. Here, we have enclosed the replies point by point to the comments and marked all the changes in red in the revised manuscript.
Because the idea of our manuscript is to study the mechanism of effect of BA and MS on LPS induced BV2 cells. According to the correlation analysis between the content and effect of BA and MS, it was found that the effective substances were 2-methoxy-5-acetoxy-fruranogermacr-1 (10)-en-6-one (compound 1), 3-acetyloxylanosta-8,24-dien-21-oic acid (compound 2), 11-keto-boswellic acid (compound 3). 3- acetyl-11-keto-bet-boswellic acid (compound 4). Then, the four compounds were further analyzed by network pharmacology, and the anti-inflammatory effects of these four compounds have been reported, and we consider verifying them in future experiments. In terms of the effects of compounds, we've reorganized the sentences in the discussion section and marked them in red. We have further described the Toll-like signaling pathways analyzed by network pharmacology and adjusted the sentences in the discussion section. We have reorganized the sentences in the abstract section. We have cited more recent research-related literatures.
Reviewer 3 Report
The manuscript in reference describes the evaluation of the anti-neuroinflammation effect and plausible mechanisms of several preparations at different proportions of boswellic acid (BA) and myrrha sesquiterpenes (MS) on BV2 cells. The manuscript has interesting results, however, some major points must be addressed prior further consideration.
The manuscript requires a detailed scrutiny regarding style and grammar, since some sentences have several language problems and make difficult to follow the ideas. An English editing service is then recommended. Introduction is very disconnected between paragraphs. I suggest to reorganize and connect the ideas in introduction section. I consider the results are poorly described. Some results require more information about importance/significance, because they are not clear in some parts. For instance, there is not clarity about the 32 selected key targets and the respective description about table 2 and figure 6. Similar situation occurs in figure 4 and 5. Similar to that of introduction, discussion is not clear. Perhaps a subdivision in several subtitles could help for such a clarity within ideas of the discussion section. Conclusions are very superficial. I consider results and discussion can allow a more elaborate and deeper conclusions. In addition, the final sentence "...which indicates that the ratio of BA and MS can be used to treat neuroinflammation caused by various causes..." is very ambiguous and is not sufficiently supported by results. Finally, and may be the most important one, in manuscript is not clear what is the aim and scope of the manuscript. I consider very important to clarify this. Additionally, I recommend to authors to test the individual compounds (mentioned in section 4.2.6.3.) to support the informatics-based analyses and network pharmacology.
Author Response
Thanks for the reviewer’s professional comments. Here, we have enclosed the replies point by point to the comments and marked all the changes in red in the revised manuscript.
1.We have corrected the language with the help of native English speakers.
We have reorganized the language and sentences in the introduction section and marked them in red. We have refined the description of the results in Table 2 and Figure 6, Figure 4 and Figure 5, with the modifications highlighted in red. We have already supplemented the abstract section, the introduction section, the discussion section and the conclusions section, and the relevant sentences have been properly adjustedand marked in red. Because this manuscript is based mainly on BV2 cell and network pharmacology to study the mechanism of anti-neuroinflammatory effects of BAand MS with different compatibility, some compounds in 4.2.6.3 are not verified, we will consider the experiment of some compounds in 4.2.6.3 in future studies.
Round 2
Reviewer 1 Report
According to the revision, most of my concerns have been clarified and also English of the manuscript has been much improved. However, still awkward expressions are observed throughout the manuscript.
ex) In Abstract (page 3), the sentence 'Frankincense and myrrha are herbal pairs commonly used traditional Chinese medicines, which have been shown to have anti-inflammatory.' should be corrected.
The manuscript should be edited again more carefully by professional native speaker(s).
Author Response
Thanks for your advice, we have edited again the language carefully with the help of native English speakers and already marked in red.
Reviewer 3 Report
The manuscript improved in quality and content. Authors addressed the main points of my previous revision. However, I insist about the detailed scrutiny about the grammar and style, despite the mentioned revision by native English speakers. In addition, understanding the focus of the manuscript, I consider it would have been interesting if the activity of standards is evaluated, in order to add more quality to the paper, since the results related to the mechanism are provided from a theoretical point of view, and a experimental validation is recommended in the same manuscript. This is not a big problem, but this manuscript version leaves the feeling of an incomplete work.
Author Response
1.Thanks for your advice, we have edited again the language carefully with the help of native English speakers and already marked in red..
2.Thank you so much for your professional and useful advice. Indeed, as you said, this manuscript will be more perfect if there are compounds activity evaluation to verify the theory. But this work is time-consuming and labor-intensive, we have planned to do this in the following work.